# Environmental pathogen surveillance in cities without universal piped wastewater infrastructure

Drew Capone[1]*, Marcia Chiluvane[2], Victoria Cumbane[2], Jack Dalton[3], David Holcomb[4], Erin Kowalsky[4], Amanda Lai[5], Elly Mataveia[2], Vanessa Monteiro[2], Gouthami Rao[4], Edna Viegas[2], Joe Brown[4]

1 Department of Environmental and Occupational Health, Indiana University, Bloomington, Indiana, United States of America, 2 Centro de Investigação e Treino em Saúde da Polana Caniço, Instituto Nacional de Saúde, Maputo, Mozambique, 3 School of Civil Engineering, University of Leeds, Leeds, United Kingdom, 4 Department of Environmental Sciences and Engineering, University of North Carolina-Chapel Hill, Chapel Hill, North Carolina, United States of America, 5 The Aquaya Institute, San Anselmo, California, United States of America

* dscapone@iu.edu

## Abstract

Although wastewater surveillance for assessing community health and well-being is now mainstream, most cities in low- and middle-income countries lack conventional wastewater services. In these settings, environmental surveillance beyond conventional wastewater offers the potential to inform public health responses, design interventions intended to reduce exposures, and to evaluate infection control programs. To explore these potential use cases, we measured pathogens, source-tracking markers, and fecal indicator bacteria in wastewater treatment plant (WWTP) influent and effluent, wastewater surface discharges, impacted river water, impacted soils, open drains, stormwater, and fecal sludges from onsite sanitation in Maputo, Mozambique. We detected a wide range of pathogens by multi-parallel RT-qPCR across all matrices, revealing a nuanced picture of pathogen flows in the city and suggesting the potential for exposures beyond those typically included in studies of sanitation and health. We developed a regression model with multiple pathogens as the dependent variable and observed lower pathogen concentrations in direct wastewater discharges (mean difference -1.4 $\log_{10}$ per liter, 95% CI: -1.7, -1.1), WWTP effluent (-0.97 $\log_{10}$, 95% CI: -1.5, -0.47), water from open drains (-2.0 $\log_{10}$, 95% CI: -2.5, -1.6), impacted river water (-3.0 $\log_{10}$, 95% CI: -3.7, -2.4), and stormwater (-4.7 $\log_{10}$, 95% CI: -7.0, -3.3) compared to WWTP influent. We further observed that a one standard deviation increase in 7-day cumulative precipitation was associated with an increase in the pathogen concentration in all matrices (0.11 $\log_{10}$, [0.04, 0.19]). Despite lower concentrations of pathogens in matrices compared to WWTP influent, frequent detection of pathogens indicates clear potential to use environmental pathogen surveillance to inform public health responses in cities lacking universal conventional wastewater, with a wide range of promising applications.

**Data availability statement:** Protocols and data used in this paper are available at a dedicated data repository at Open Science Framework (OSF.io). Permanent link: https://osf.io/h5epg/.

**Funding:** This study was funded by the Bill & Melinda Gates Foundation (www.gatesfoundation.org) Grant (OPP1137224 to JB). DC was supported in part by an NIH T32 Fellowship (5T32ES007018-44). The funders had no role in study design, data collection and analysis, decision to publish, or preparation of the manuscript.

**Competing interests:** The authors have declared that no competing interests exist.

## Introduction

Wastewater surveillance or wastewater-based epidemiology (WBE) has become a mainstream approach for detection and quantification of pathogens of public health interest. WBE encompasses a wide range of tools, approaches, and applications across scales including variant analysis, estimation of burdens and infection trends, and generating information about spatial and temporal patterns of transmission to better target interventions or control measures where they can be most effective. The potential for these methods to contribute to public health gains may be greatest where they currently are the least developed and least scalable: in low- and middle-income countries (LMICs) with a high disease burden and poor sanitation coverage [1–3]. LMIC settings present challenges to emerging consensus methods for wastewater surveillance and data interpretation, representing use cases that differ in important ways from those in high-income countries [3].

Pathogens that are shed in feces or urine are the primary candidates for WBE. This includes enteric pathogens (e.g., *Shigella*, *Giardia*, norovirus) which are transmitted via the fecal-oral route, as well as respiratory (e.g., SARS-CoV-2, influenza) and vector-borne pathogens (e.g., *Plasmodium*, zika) that can be shed in but are not transmitted via feces [4–7]. In urban areas of high income countries, feces in wastewater is conveyed from nearly all individuals living in a sewershed to a centralized treatment plant. However, at the most basic level, cities and towns in LMICs may be wholly unserved by conventional, networked wastewater systems or they may be partly served by conventional and onsite systems of various types [8]. Drainage networks and urban surface waters may receive high quantities of fecal wastes in these settings. Due to the prevalence of onsite systems such as pit latrines and septic tanks, representative sampling of fecal sludges may be needed [8–10].

The expanded set of matrices beyond conventional wastewater included in monitoring efforts has given rise to the term *environmental surveillance* (ES) as distinct from WBE in piped sewerage systems. While ES draws on similar sampling and analysis techniques, its use cases in cities with limited sewer coverage may be different. Poorly defined catchment areas, decentralized sanitation systems, and uncertain spatial and temporal representativeness of matrices such as open drains or fecal sludges may limit the feasibility of estimating population-level infection burdens. Challenges would likely vary across cities given the substantial heterogeneity in sanitation and drainage infrastructure in low-income urban settings. Analytical capacity may also constrain what can be done with samples in local laboratories, and these challenges may become more acute with reductions in global health aid that bolster laboratory facilities, equipment, and support [11].

Despite these challenges, ES offers a wide range of public health applications in LMICs. These include some of those now used in WBE: identification of novel or emerging pathogens, monitoring temporal trends in environmental matrices, evaluating vaccine or mass drug administration programs, assessing the effectiveness of sanitation or infrastructure interventions, and tracking the emergence and dissemination of antimicrobial resistance (AMR) [3,7,9,12–15]. While such work in high-income settings with good sanitation infrastructure may not be directly relevant to exposures,

an application of ES in LMICs is the identification of pathogen hazards associated with uncontained excreta: that is, detection of a pathogen in wastewater or in an open drain attains new relevance if there are opportunities for people to come into contact with these waste streams [14–18].

Empirical data describing the presence, concentration, and variance of enteric pathogens across matrices provides the foundation for developing environmental surveillance systems in cities lacking conventional wastewater infrastructure. Understanding variance and prevalence for specific pathogens informs sample size calculations and would allow future surveillance efforts to meet minimum probability thresholds for detecting pathogen signals with statistical confidence. Additionally, estimating the proportion of pathogens released into the environment through uncontained excreta helps quantify environmental hazards, support exposure assessment, and prioritize public health interventions most likely to reduce transmission risks [19,20].

In this study, we sought to examine fecal waste streams in a large city with limited sanitation infrastructure, comparing quantitative data for a wide range of molecular targets in each matrix across two seasons. Our primary goal was to assess mean differences in molecular target concentrations relative to wastewater influent, in order to better understand the potential of alternative environmental matrices for ES. Our other goals were to assess presence, concentration, and variance of enteric pathogens in sample types to inform sample size calculations for potential future use cases, to estimate the fraction of pathogens associated with uncontained excreta [21], and to understand the impact of seasonal variation [22,23] on these signals.

## Methods

### Ethics statement

The Maputo Sanitation trial, which included this study, was approved by Comité Nacional de Bioética para a Saúde, Ministério da Saúde de Moçambique (FWA#: 00003139, IRB00002657, 326/CNBS/21; approved: 15 June 2021) and University of North Carolina at Chapel Hill Ethics Committee (IRB#: 21–1119; approved 19 August 2021) and was prospectively registered with ISRCTN on 16 March 2022 (https://doi.org/10.1186/ISRCTN86084138) (S1 Text) [24]. We received written approval for sampling from the Maputo Municipal Council's Division of Urban Infrastructure.

### Study site

We conducted two cross-sectional studies in Maputo, Mozambique: one in the rainy season and one in the dry season. Maputo is a rapidly growing metropolitan area, with water, sanitation, and hygiene (WASH) challenges common to low- and middle-income countries [25–27]. The Maputo metropolitan area has a population of over 2.7 million, but fewer than 150,000 of Maputo's residents are connected to a sewer system, leading to significant strain on the existing infrastructure due to rapid population growth [28].

The sewer network in Maputo consists of two systems [29]. The first system was built in the 1940s as a drainage system but now functions as a combined sewer system and discharges directly into Maputo Bay (hereafter referred to as the combined system, light blue area in Figs 1 and S1). The second system, built in the 1980s, consists of sewer lines, a WWTP and two pumping stations (hereafter referred to as the separated sewer system, red area in Fig 1). The pumping stations were designed to pump wastewater from the combined system into the newer separated system (i.e., from blue to red in Fig 1), but the pumping stations have been inoperable since the year 2000. Fecal sludge and industrial waste are co-treated with wastewater at the WWTP, which is a series of anaerobic and facultative stabilization ponds in parallel. Effluent from the treatment plant is discharged to the Infulene River.

### Sampling locations

We sought to sample from environmental matrices visualized on Maputo's SFD in the public domain [30] (i.e., outside the domestic living environment) during both the rainy (October-April) and dry (May-September) seasons. These matrices include wastewater influent from the separated system and WWTP effluent, wastewater surface discharge from the

**Fig 1. Sanitation coverage in Maputo.** Households in the blue and red areas are served by piped sewer systems. Households in the yellow area use onsite sanitation. Basemap data are derived from OpenStreetMap contributors (https://www.openstreetmap.org/copyright) and were accessed via Esri. OpenStreetMap data are licensed under the Open Database License (ODbL). The map was created by the authors using ArcGIS. Coordinate system: WGS 1984 Web Mercator Auxiliary Sphere.

combined system at the point of discharge into Maputo Bay, water from open drains, river water, stormwater, soil from farms adjacent to the WWTP, soil adjacent to public waste receptacles, and fecal sludge from onsite sanitation systems following transport and discharge to the WWTP (S2 Fig). We used a purposive approach, as this enabled us to select the most appropriate sampling locations to achieve our objectives. First, we met with municipal technicians responsible for the sewer network and drains in Maputo, who provided advice on accessing sites. Next, we combined our background knowledge from living and working in Maputo with data from gray literature and publicly accessible satellite imagery to identify possible sampling locations [25,29,31,32]. We then visited potential locations to assess the safety and feasibility of sampling.

Where only one sampling location existed (e.g., WWTP influent and effluent) we aimed to collect multiple samples in each season. For matrices where multiple locations of the same matrix existed (e.g., open drains) we prioritized geographic diversity. Stormwater – which we defined as pooled standing water with a surface area ≥20 m$^2$ – was collected following rainfall events via a convenience sample. This intermittent flooding was highly localized and ephemeral following

rainfall. Given that the Infulene River receives water from the Maputo's largest open drain and WWTP effluent, we collected river water upstream and downstream of these discharges. When feasible, we collected matched samples at the same locations during the wet (January-March) and dry (July-August) seasons in 2022; no stormwater samples were collected in the dry season. This approach allowed us to include all matrix types, as well as capture the spatial and temporal heterogeneity present in these matrices.

We collected 96 water samples, including six samples of stormwater, 34 from open drains, 24 of impacted river water, 10 of WWTP influent (from the separated sewer system), 10 of WWTP effluent, and 12 of wastewater surface discharge into Maputo Bay (from the combined sewer system). At the WWTP, we collected 46 grab samples of fecal sludge where de-sludging trucks discharge latrine waste; 24 were randomly selected for analysis using molecular methods. We also collected soil samples, including 62 adjacent to public waste receptacles and 22 from subsistence farms adjacent to the WWTP.

## Water sampling and concentration

We collected and performed primary concentration of water samples (i.e., wastewater, river water, open drain water) using the bag-mediated filtration system (BMFS, Scientific Methods, Granger, Indiana) (S3 Fig and S2 Text) [33–36]. Bags were dragged through the water source to collect approximately six liters of water, which was gravity filtered through a positively charged ViroCap filter with a pore size of 2–3 μm (Scientific Methods, Granger, Indiana) [37]. We used the standard secondary concentration procedure for BMFS from Zhou *et al*. 2018 to eluate microbes from the ViroCap filter, concentrated eluant via skimmed milk flocculation, and resuspend the pellet in 6 mL of sterile distilled water (S2 Text) [35]. A 200 mL grab sample was also collected using a Whirl-Pak bag, which was only used to culture fecal indicator bacteria (Nasco, Pleasant Prairie, Wisconsin).

We did not use BMFS to concentrate fecal sludge in the field as we expected that the solids content would have rendered BMFS unfeasible due to filter clogging. Instead, we collected 50 mL grab samples of sludge from the point of discharge (S4 Fig) using the Sludge Nabber sampling device (Nasco, Pleasant Prairie, Wisconsin) into sterile 50 mL centrifuge tubes (VWR, Radnor, PA) [9].

## Soil sampling

We collected soil samples from subsistence farms adjacent to the WWTP and from public waste receptacles (S5 Fig). We walked transects along the western and eastern sides of the WWTP and only collected soil at subsistence plots where the owner was present and verbally assented to collection. We collected a convenience sample of soil adjacent to public waste receptacles across the city. Sampling of soil at waste receptacles was standardized by collecting at the center of the bin on the pedestrian side of the street. An approximately 10x10x1 cm volume of soil was homogenized using a sterilized lab scoopula and collected directly into three 2mL cryovials and one 5mL cryovial [38,39].

## Culture methods

We cultured *E. coli* using IDEXX Quanti-tray/2000 system (IDEXX, Westbrook, Maine). Collected water grab samples and 1g of soil samples (wet weight) were diluted in ten-fold serial dilutions, ranging from 1:10–1:10⁶, combined with Colilert-18 reagent, incubated at 37°C for 18 hours, and fluorescent wells visually counted in a UV reading chamber (IDEXX, Westbrook, Maine) (S2 Text). Approximately 5g of soil was dried using the microwave oven method to determine the moisture content [40]. We included one negative control (i.e., distilled water) each day.

## Molecular analysis

Sample aliquots were frozen at -20°C for 2–6 months and then shipped from Centro de Investigação e Treino em Saúde da Polana Caniço (CISPOC) in Maputo, Mozambique to the University of North Carolina at Chapel Hill on

dry ice with temperature monitoring where they were stored at -80°C. We extracted nucleic acids from either 200 µL of resuspended pellet or 200 µL of fecal sludge using the Qiagen QIAamp 96 Virus QIAcube HT Kit, which was automated on the QiaCube (Qiagen, Hilden, Germany). Molecular analysis of soil samples is reported separately (Dalton *et al.*, in prep). Vaccine-derived bovine herpes virus 1 (BHV, a DNA-genome virus) and bovine respiratory syncytial virus (BRSV, an RNA-genome virus) were used as extraction positive controls, while molecular water was used for extraction negative controls [41]. We included at least one negative extraction control each day of extractions.

We analyzed nucleic acids for enteric pathogen genes and microbial source tracking (MST) markers using reverse transcription quantitative PCR (RT-qPCR) via a custom TaqMan Array Card (TAC) (S2 Text and S1–S3 Tables). The targets on the TAC included the viruses adenovirus 40/41, astrovirus, norovirus GI/GII, rotavirus, sapovirus, and SARS-CoV-2; the bacteria *Aeromonas* spp., *Campylobacter jejuni/coli*, *E. coli* O157, *Clostridioides difficile* toxin B, enteroaggregative *E. coli* (EAEC), Shiga toxin-producing *E. coli* (STEC), enteropathogenic *E. coli* (EPEC), enterotoxigenic *E. coli* (ETEC), *Helicobacter pylori*; *Plesiomonas shigelloides*, *Shigella* spp., *Salmonella enterica*, *Vibrio* spp., and *Yersinia enterocolitica*; the protozoan parasites *Cryptosporidium* spp., *Entamoeba histolytica, and Giardia* spp.; and the soil transmitted helminths (STH) *Ascaris lumbricoides*, *Trichuris trichiura*, *Ancylostoma duodenale*, *Necator americanus*, and *Strongyloides stercolaris* [42]. MST markers included human mtDNA [43], canine mtDNA [44], poultry mtDNA [45], companion animal-associated *Toxocara* spp. [46], and avian 16S rRNA [45]. The class 1 integron-integrase gene (*intI1*) was included as a proxy for potential antimicrobial resistance [47]. We assayed nine replicates of a ten-fold dilution series of synthetic plasmid positive controls ($10^8$-$10^{-1}$ gene copies (gc) per µL template) to construct standard curves for quantitative estimates of gene copy densities. Additionally, we assayed seven series of low-concentration positive controls ($10^{-1}$, $10^{-0.5}$, $10^0$, $10^{0.5}$, $10^1$, $10^2$ gc/µL) to determine assay limits of detection (S2 Text) [48,49]. Positive control material concentrations were determined using dPCR on a QIACuity 4 system (Qiagen, Hilden, Germany).

## Data analysis

We generated summary statistics using R version 4.2.3 (R Foundation for Statistical Computing, Vienna, Austria). We used multiple Bayesian censored regression (*brms* package [50]) models (S2 Text), where dependent variables included individual pathogens and molecular targets, as well as multiple target and multiple pathogens [23,24,50,51]. We used a categorical variable representing the sample matrix as the independent variable to estimate the mean differences in target concentration. The dependent variable was either (A) the concentration of an individual target in $log_{10}$ transformed gene copies per liter or (B) log10 transformed concentrations for multiple targets (i.e., all pathogens, fecal source tracking markers, bacterial pathogens, protozoan pathogens, or viral pathogens). The sample-specific LODs were included in the model for non-detects, allowing them to contribute to the likelihood as a probability range (from zero to the LOD) using the cumulative distribution function. Regression models also included mean-centered and standard deviation-scaled (i.e., z-score transformed) 7-day cumulative precipitation from the Climate Hazards Group InfraRed Precipitation with Station data (CHIRPS) [52] and daily mean temperature from the National Centers for Environmental Prediction (NCEP) Climate Forecast System (CFS) standardized in the same manner as precipitation [53]. We included random intercepts and matrix-specific random slopes by molecular target. Regression models run for individual targets did not include the random effect for target and models run for specific matrices did not include the random effect for matrix.

We estimated the fraction of pathogens in each matrix that were unsafely managed using the Excreta Flow Diagram (SFD) previously developed for Maputo (S3 Text) [25]. This fraction was overlayed onto our empirical data to visualize pathogen hazards associated with the matrices assessed. In addition, we ran simulations in R to estimate the sample size necessary to determine significant reductions in target concentrations (80% power, α = 0.05) for an analysis of five commonly detected targets (S4 Text).

## Results

### Fecal indicator bacteria

We observed substantial heterogeneity in *E. coli* concentrations across environmental matrices (Fig 2). WWTP influent from the separated sewer system (mean = 7.3 $\log_{10}$ most probable number [MPN] per 100 mL, standard deviation = 0.32 $\log_{10}$), fecal sludge disposed at the WWTP (mean = 6.4 $\log_{10}$, sd = 0.48 $\log_{10}$), wastewater surface discharge into Maputo Bay from the combined sewer system (mean = 6.4 $\log_{10}$ MPN/100 mL, sd = 0.47 $\log_{10}$), and WWTP effluent (mean = 6.0 $\log_{10}$ MPN/100 mL, sd = 0.34 $\log_{10}$) had the highest concentrations of *E. coli* and the lowest variance. Water from open drains (mean = 5.4 $\log_{10}$ MPN/100 mL, sd = 0.96 $\log_{10}$), soil adjacent to public waste receptacles (mean = 4.0 $\log_{10}$ MPN per dry gram, sd = 1.3 $\log_{10}$), river water (mean = 3.7 $\log_{10}$ MPN/100 mL, sd = 1.2 $\log_{10}$), and stormwater (mean = 3.7 $\log_{10}$ MPN/100 mL, sd = 1.5 $\log_{10}$) had lower concentrations of *E. coli* but substantially higher variance. *E. coli* concentrations in river water were higher downstream of the WWTP (mean = 5.1 $\log_{10}$ MPN per 100 mL, sd = 0.34 $\log_{10}$) compared to upstream (mean = 3.1 $\log_{10}$ MPN per 100 mL, sd = 0.79 $\log_{10}$).

### Fecal source tracking markers

We most frequently detected the human-origin source tracking marker human mtDNA (86%, 104/121), followed by poultry mtDNA (24%, 29/121), canine mtDNA (8.3%, 10/121), and avian 16S rRNA (4.1%, 5/121). We did not detect *Toxocara*

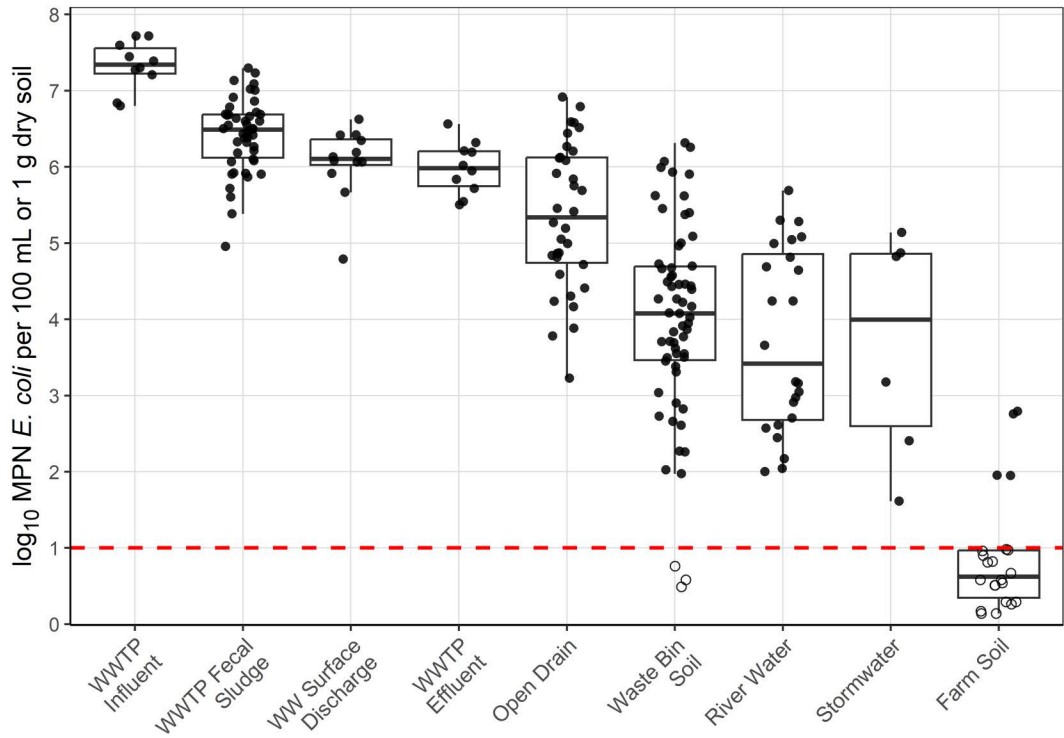

**Fig 2.** *E. coli* **concentrations in environmental matrices.** Dots below the dashed red line, which represents the lower limit of detection, are non-detects and have been randomly imputed from 0 to the LOD for visualization purposes and are shown as unfilled circles. Abbreviations: wastewater treatment plant (WWTP), wastewater (WW).

spp. (0%, 0/121), which are helminths associated with dogs and cats (Table 1). Compared to WWTP influent from the separated sewer system, fecal sludge collected at the WWTP had significantly higher concentrations of the multi-target outcome of fecal source tracking markers (1.6 $\log_{10}$ mean difference in gene copies per liter, [95% credible interval: 0.50, 2.4]). Concentrations of the fecal source tracking markers were not significantly different in wastewater surface discharge from the combined system (-1.4 $\log_{10}$, [-4.3, 0.14]), WWTP effluent (-0.88 $\log_{10}$, [-2.3, 0.27]), water from open drains (-0.54 $\log_{10}$, [-2.4, 1.5]), river water (-2.1 $\log_{10}$, [-4.4, 0.08]), or stormwater (-0.62 $\log_{10}$, [-3.6, 2.3]) compared to WWTP influent.

### Enteric pathogens

We frequently detected nucleic acids from enteric bacteria, viruses, and protozoa, but rarely from soil transmitted helminths (Table 1). We detected *Aeromonas* spp. most frequently (93%, 113/121), followed by norovirus GI/GII (65%, 78/121), *Cryptosporidium* spp. (62%, 75/121), EAEC (61%, 74/121), *Vibrio* spp. (61%, 74/121), EPEC (60%, 72/121), adenovirus 40/41 (55%, 67/121), *Giardia* spp. (54%, 65/121), astrovirus (52%, 63/121), and rotavirus (52%, 63/121).

Concentrations of the multi-pathogen outcome were not significantly different in fecal sludge collected at the WWTP (0.19 $\log_{10}$ mean difference in gene copies per liter, [-0.45, 0.83]) compared to WWTP influent (Table 2). We observed lower concentrations of pathogens in wastewater surface discharge (-1.4 $\log_{10}$, [-1.7, -1.1]), WWTP effluent (-0.97 $\log_{10}$, [-1.5, -0.47]), water from open drains (-2.0 $\log_{10}$, [-2.5, -1.6]), river water (-3.0 $\log_{10}$, [-3.7, -2.4), and stormwater (-4.7 $\log_{10}$, [-7.0, -3.3]) compared to WWTP influent. The directionality of the point estimate for the difference in the pathogen estimate was generally consistent with the directionality of point estimates for by pathogen class (Table 2).

Concentrations of the multi-bacterial pathogen outcome were significantly lower in fecal sludge collected at the WWTP (-0.80 $\log_{10}$, [-1.3, -0.25]), wastewater surface discharge (-1.4 $\log_{10}$, [-1.9, -1.1]), WWTP effluent (-1.5 $\log_{10}$, [-1.9, -1.2]), water from open drains (-2.1 $\log_{10}$, [-2.6, -1.6]), river water (-3.5 $\log_{10}$, [-4.3, -2.8]), and stormwater (-4.6 $\log_{10}$, [-7.2, -3.2]) compared to WWTP influent.

Concentrations of the multi-viral pathogen estimate were not significantly different in fecal sludge collected at the WWTP (0.68 $\log_{10}$, [-0.93, 2.4]) and WWTP effluent (-0.27 $\log_{10}$, [-1.7, 0.9]) compared to WWTP influent (Table 2). Wastewater surface discharge (-1.3 $\log_{10}$, [-3.0, -0.29]), water from open drains (-2.2 $\log_{10}$, [-2.9, -1.3), river water (-3.0 $\log_{10}$, [-4.0, -2.1), and stormwater (-7.4 $\log_{10}$, [-15, -3.8]) had significantly lower concentrations of the viral pathogen estimate compared to WWTP influent.

Water from open drains (-2.2 $\log_{10}$, [-5.2, 0.0]) had significantly lower concentrations of the multi-protozoan pathogen estimate compared to wastewater influent. Concentration of the protozoan pathogen were not significantly different in fecal sludge collected at the WWTP (0.57 $\log_{10}$, [-2.4, 3.5]), wastewater surface discharge (-1.5 $\log_{10}$, 95% CI: -3.1, 0.03), WWTP effluent (-0.23 $\log_{10}$, [-2.3, 2.0]), river water (-2.5 $\log_{10}$, [-7.9, 0.90]), and stormwater (-2.9 $\log_{10}$, [-11, 2.7]) compared to WWTP influent (Table 2).

### Precipitation and temperature

Overall, increasing precipitation and temperature were associated with small but statistically significant increases in the targets we evaluated. A one standard deviation (i.e., z-score) increase in 7-day cumulative precipitation (millimeters) was associated with an increased pathogen concentration (0.11 $\log_{10}$, [0.04, 0.19]) (Table 2). The positive relationship between precipitation and pathogen concentration was greatest for bacterial pathogens (0.17 $\log_{10}$, [0.08, 0.26]), but was not significant for viruses and protozoan pathogens. Most point estimates for individual targets indicate a positive association between precipitation and target concentrations, but few were statistically significant (Table 2). Culturable *E. coli* was positively associated with increasing rainfall (0.18 $\log_{10}$, [0.04, 0.33]). The human mtDNA fecal source tracking marker (FST) was also positively associated with increasing rainfall (0.19 $\log_{10}$, [0.04, 0.35]). *Entamoeba histolytica* was the only target that was negatively associated with increasing rainfall (-0.88 $\log_{10}$, [-1.7, -0.17]).

**Table 1. Prevalence and $\log_{10}$ transformed concentration of targets in environmental matrices per liter volume followed by $\log_{10}$ transformed standard deviations in parentheses.**

| Individual Target | WWTP Influent | | Fecal Sludge (from WWTP) | | WW Surface Discharge | | WWTP Effluent | | Open Drain | | River Water | | Stormwater | |
|---|---|---|---|---|---|---|---|---|---|---|---|---|---|---|
| | Detects | Mean (sd) | Detects | Mean (sd) | Detects | Mean (sd) | Detects | Mean (sd) | Detects | Mean (sd) | Detects | Mean (sd) | Detects | Mean (sd) |
| 16S rRNA | 100% (10/10) | 12.7 (0.62) | 100% (24/24) | 13.9 (0.42) | 100% (12/12) | 11.8 (0.61) | 100% (10/10) | 12.3 (0.75) | 100% (34/34) | 11.3 (0.69) | 100% (24/24) | 10.7 (0.55) | 100% (6/6) | 11.1 (0.44) |
| A. duodenale | 0% (0/10) | <LOD | 8.3% (2/24) | <LOD | 0% (0/12) | <LOD | 0% (0/10) | <LOD | 0% (0/34) | <LOD | 0% (0/24) | <LOD | 0% (0/6) | <LOD |
| A. lumbricoides | 0% (0/10) | <LOD | 0 (0/24) | <LOD | 0% (0/12) | <LOD | 0% (0/10) | <LOD | 0% (0/34) | <LOD | 0% (0/24) | <LOD | 0% (0/6) | <LOD |
| Adenovirus 40/41 | 100% (10/10) | 6.2 (0.65) | 25% (6/24) | <LOD | 83% (10/12) | 5 (2.0) | 100% (10/10) | 6.1 (0.29) | 62% (21/34) | 3.8 (1.6) | 46% (11/24) | <LOD | 0% (0/6) | <LOD |
| Aeromonas | 100% (10/10) | 7.8 (0.57) | 71% (17/24) | 6.4 (2.1) | 100% (12/12) | 7 (0.77) | 100% (10/10) | 6.4 (0.78) | 100% (34/34) | 7.1 (0.96) | 100% (24/24) | 5.4 (1.1) | 83% (5/6) | 4.4 (1.2) |
| Astrovirus | 100% (10/10) | 6.9 (0.38) | 54% (13/24) | 6.3 (2.5) | 92% (11/12) | 5.6 (1.1) | 100% (10/10) | 7 (0.32) | 41% (14/34) | 3.3 (2.2) | 25% (6/24) | <LOD | 0% (0/6) | <LOD |
| Avian 16S | 0% (0/10) | <LOD | 0% (0/24) | <LOD | 0% (0/12) | <LOD | 0% (0/10) | <LOD | 5.9% (2/34) | <LOD | 0% (0/24) | <LOD | 50% (3/6) | 3.7 (1.3) |
| C. difficile | 70% (7/10) | 3.5 (1.3) | 0% (0/24) | <LOD | 8.3% (1/12) | <LOD | 10% (1/10) | <LOD | 5.9% (2/34) | <LOD | 0% (0/24) | <LOD | 0% (0/6) | <LOD |
| C. jejuni/coli | 90% (9/10) | 4.6 (0.88) | 4.2% (1/24) | <LOD | 67% (8/12) | 3.2 (1) | 20% (2/10) | <LOD | 18% (6/34) | <LOD | 4.2% (1/24) | <LOD | 0% (0/6) | <LOD |
| Canine mtDNA | 20% (2/10) | <LOD | 8.3% (2/24) | <LOD | 0% (0/12) | <LOD | 10% (1/10) | <LOD | 18% (6/34) | <LOD | 4.2% (1/24) | <LOD | 17% (1/6) | <LOD |
| Cryptosporidium | 70% (7/10) | 3.9 (1.1) | 50% (12/24) | 5 (2.4) | 50% (6/12) | 3.5 (2.0) | 100% (10/10) | 5.4 (0.56) | 56% (19/34) | <LOD | 83% (20/24) | 3.7 (1.7) | 83% (5/6) | 5.6 (2.4) |
| E. coli O157:H7 | 100% (10/10) | 4.6 (0.28) | 4.2% (1/24) | <LOD | 8.3% (1/12) | <LOD | 10% (1/10) | <LOD | 15% (5/34) | <LOD | 4.2% (1/24) | <LOD | 0% (0/6) | <LOD |
| E. histolytica | 90% (9/10) | 5.5 (1.28) | 8.3 (2/24) | <LOD | 42% (5/12) | <LOD | 80% (8/10) | 4.4 (1.4) | 12% (4/34) | <LOD | 8.3% (2/24) | <LOD | 0% (0/6) | <LOD |
| EAEC | 100% (10/10) | 6.9 (0.3) | 38% (9/24) | <LOD | 100% (12/12) | 5.9 (0.71) | 100% (10/10) | 5.6 (0.28) | 79% (27/34) | 4.9 (1.7) | 17% (4/24) | <LOD | 17% (1/6) | <LOD |
| EPEC | 100% (10/10) | 6.7 (0.27) | 46% (11/24) | <LOD | 100% (12/12) | 5.5 (0.75) | 100% (10/10) | 5.1 (0.47) | 62% (21/34) | 3.8 (2.0) | 21% (5/24) | <LOD | 33% (2/6) | <LOD |
| ETEC | 100% (10/10) | 6.4 (0.46) | 33% (8/24) | <LOD | 50% (6/12) | 3.5 (2.0) | 100% (10/10) | 5.4 (0.53) | 44% (15/34) | 3 (2.0) | 17% (4/24) | <LOD | 0% (0/6) | <LOD |
| Giardia | 100% (10/10) | 5.8 (0.82) | 83% (20/24) | 6.5 (1.8) | 92% (11/12) | 4.4 (0.92) | 100% (10/10) | 4.7 (0.67) | 32% (11/34) | <LOD | 8.3% (2/24) | <LOD | 17% (1/6) | <LOD |
| H. pylori | 0% (0/10) | <LOD | 0% (0/24) | <LOD | 0% (0/12) | <LOD | 0% (0/10) | <LOD | 0% (0/34) | <LOD | 0% (0/24) | <LOD | 0% (0/6) | <LOD |
| HIV | 0% (0/10) | <LOD | 0% (0/24) | <LOD | 0% (0/12) | <LOD | 0% (0/10) | <LOD | 0% (0/34) | <LOD | 0% (0/24) | <LOD | 0% (0/6) | <LOD |
| human mtDNA | 100% (10/10) | 7.1 (0.64) | 100% (24/24) | 8.6 (0.64) | 100% (12/12) | 5.6 (0.51) | 100% (10/10) | 5.6 (0.36) | 85% (29/34) | 4.4 (1.5) | 54% (13/24) | 3.2 (1.6) | 83% (5/6) | 4.1 (0.67) |

*(Continued)*

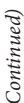

Table 1. (Continued)

| Individual Target | WWTP Influent | | Fecal Sludge (from WWTP) | | WW Surface Discharge | | WWTP Effluent | | Open Drain | | River Water | | Stormwater | |
|---|---|---|---|---|---|---|---|---|---|---|---|---|---|---|
| | Detects | Mean (sd) | Detects | Mean (sd) | Detects | Mean (sd) | Detects | Mean (sd) | Detects | Mean (sd) | Detects | Mean (sd) | Detects | Mean (sd) |
| intl1 | 100% (10/10) | 10 (0.30) | 100% (24/24) | 10 (0.51) | 100% (12/12) | 9.1 (0.51) | 100% (10/10) | 9.8 (0.24) | 100% (34/34) | 8.6 (0.98) | 100% (24/24) | 7.1 (1.1) | 100% (6/6) | 7.5 (0.47) |
| *Leptospira* | 0% (0/10) | <LOD | 4.2% (1/24) | <LOD | 0% (0/12) | <LOD | 0% (0/10) | <LOD | 0% (0/34) | <LOD | 0% (0/24) | <LOD | 0% (0/6) | <LOD |
| *M. tuberculosis* | 0% (0/10) | <LOD | 0% (0/24) | <LOD | 0% (0/12) | <LOD | 10% (1/10) | <LOD | 0% (0/34) | <LOD | 0% (0/24) | <LOD | 0% (0/6) | <LOD |
| *N. americanus* | 0% (0/10) | <LOD | 4.2% (1/24) | <LOD | 0% (0/12) | <LOD | 0% (0/10) | <LOD | 0% (0/34) | <LOD | 0% (0/24) | <LOD | 0% (0/6) | <LOD |
| Norovirus | 100% (10/10) | 5.2 (0.62) | 75% (18/24) | 6.2 (2.5) | 100% (12/12) | 4.9 (0.66) | 100% (10/10) | 5.7 (0.16) | 53% (18/34) | 3.1 (1.4) | 38% (9/24) | <LOD | 0% (0/6) | <LOD |
| Poultry mtDNA | 60% (6/10) | 3.2 (1.2) | 29% (7/24) | <LOD | 25% (3/12) | <LOD | 30% (3/10) | <LOD | 32% (11/34) | <LOD | 4.2% (1/24) | <LOD | 17% (1/6) | <LOD |
| Rotavirus | 90% (9/10) | 5.8 (1.8) | 38% (9/24) | <LOD | 83% (10/12) | 4.8 (1.9) | 100% (10/10) | 6.9 (0.24) | 50% (17/34) | 3.5 (2.0) | 33% (8/24) | <LOD | 17% (1/6) | <LOD |
| *Salmonella* | 60% (6/10) | 3.2 (1.3) | 0% (0/24) | <LOD | 33% (4/12) | <LOD | 0.2 (2/10) | <LOD | 0% (0/34) | <LOD | 0% (0/24) | <LOD | 0% (0/6) | <LOD |
| Sapovirus | 90% (9/10) | 6.4 (1.7) | 54% (13/24) | 6.3 (2.3) | 83% (10/12) | 5.4 (1.6) | 100% (10/10) | 7 (0.38) | 32% (11/34) | <LOD | 8.3% (2/24) | <LOD | 17% (1/6) | <LOD |
| SARS-CoV-2 | 40% (4/10) | 3.7 (1.8) | 13% (3/24) | <LOD | 0% (0/12) | <LOD | 0.1 (1/10) | <LOD | 2.9% (1/34) | <LOD | 0% (0/24) | <LOD | 0% (0/6) | <LOD |
| *Shigella*/ EIEC | 100% (10/10) | 6 (0.36) | 17% (4/24) | <LOD | 92% (11/12) | 4.1 (0.88) | 90% (9/10) | 4.2 (1.3) | 44% (15/34) | <LOD | 4.2% (1/24) | <LOD | 0% (0/6) | <LOD |
| STEC | 80% (8/10) | 4.4 (1.7) | 4% (1/24) | <LOD | 25% (3/12) | <LOD | 30% (3/10) | <LOD | 15% (5/34) | <LOD | 0% (0/24) | <LOD | 0% (0/6) | <LOD |
| *T. trichiura* | 10% (1/10) | <LOD | 0% (0/24) | <LOD | 0% (0/12) | <LOD | 0% (0/10) | <LOD | 18% (6/34) | <LOD | 13% (3/24) | <LOD | 0% (0/6) | <LOD |
| *Toxocara* spp. | 0% (0/10) | <LOD | 0 (0/24) | <LOD | 0% (0/12) | <LOD | 0% (0/10) | <LOD | 0% (0/34) | <LOD | 0 (0/24) | <LOD | 0% (0/6) | <LOD |
| *Vibrio* spp. | 100% (10/10) | 5.9 (0.29) | 13% (3/24) | <LOD | 83% (10/12) | 4.2 (1.7) | 70% (7/10) | 3.5 (1.5) | 82% (28/34) | 4.8 (1.9) | 54% (13/24) | 3.2 (1.4) | 83% (5/6) | 4.5 (1.0) |
| Zika | 10% (1/10) | <LOD | 33% (8/24) | <LOD | 0% (0/12) | <LOD | 0% (0/10) | <LOD | 8.8% (3/34) | <LOD | 0% (0/24) | <LOD | 0% (0/6) | <LOD |

Abbreviations: enteroaggregative *E. coli* (EAEC), enteropathogenic *E. coli* (EPEC), enterotoxigenic *E. coli* (ETEC), shiga-toxin producing *E. coli* (STEC), enteroinvasive *E. coli* (EIEC), mitochondrial DNA (mtDNA), Class 1 Integron-Integrase Gene (intl1), standard deviation (sd).

**Table 2. Mean difference in log₁₀ transformed target concentration estimates in gene copies per liter between environmental matrices relative to the reference, followed by 95% credible intervals in parentheses.**

| Independent Variable | Fecal Sludge (from WWTP) | WW Surface Discharge | WWTP Effluent | Open Drain | River Water | Stormwater | Precipitation | Temperature |
|---|---|---|---|---|---|---|---|---|
| **Reference Variable** | WWTP Influent | | | | | | 1 standard deviation increase | 1 standard deviation increase |
| **Multi-target models** | | | | | | | | |
| Pathogens | 0.19 (-0.45, 0.83) | -1.4 (-1.7, -1.1) | -0.97 (-1.5, -0.47) | -2.0 (-2.5, -1.6) | -3.0 (-3.7, -2.4) | -4.7 (-7.0, -3.3) | 0.11 (0.04, 0.19) | 0.09 (0.03, 0.15) |
| Fecal source tracking markers | 1.6 (0.50, 2.4) | -1.4 (-4.3, 0.14) | -0.88 (-2.3, 0.27) | -0.54 (-2.4, 1.5) | -2.1 (-4.4, 0.08) | -0.62 (-3.6, 2.3) | 0.12 (-0.01, 0.25) | -0.16 (-0.30, -0.04) |
| Protozoan pathogens | 0.57 (-2.4, 3.5) | -1.5 (-3.1, 0.03) | -0.23 (-2.4, 2.0) | -2.2 (-5.2, 0.0) | -2.5 (-7.9, 0.90) | -2.9 (-11, 2.7) | 0.02 (-0.13, 0.17) | 0.03 (-0.11, 0.16) |
| Bacterial pathogens | -0.80 (-1.3, -0.25) | -1.5 (-1.9, -1.1) | -1.5 (-1.9, -1.2) | -2.1 (-2.6, -1.6) | -3.5 (-4.3, -2.8) | -4.6 (-7.2, -3.2) | 0.17 (0.08, 0.26) | 0.04 (-0.05, 0.12) |
| Viral pathogens | 0.68 (-0.93, 2.4) | -1.3 (-3.0, -0.29) | -0.27 (-1.7, 0.86) | -2.2 (-2.9, -1.3) | -3.0 (-4.0, -2.1) | -7.4 (-15, -3.8) | 0.04 (-0.13, 0.22) | 0.19 (0.04, 0.33) |
| **Individual target models** | | | | | | | | |
| culturable E. coli | -1.0 (-1.8, -0.30) | -1.3 (-2.2, -0.40) | -1.3 (-2.3, -0.45) | -2.0 (-2.8, -1.3) | -3.6 (-4.4, -2.8) | -4.0 (-5.1, -2.9) | 0.18 (0.04, 0.33) | 0.01 (-0.14, 0.16) |
| human mtDNA | 1.4 (0.85, 2.0) | -1.5 (-2.1, -0.82) | -1.5 (-2.2, -0.81) | -2.5 (-3.1, -2.0) | -3.3 (-3.9, -2.7) | -3.3 (-4.2, -2.4) | 0.19 (0.04, 0.35) | -0.05 (-0.19, 0.09) |
| intl1 | -0.10 (-0.71, 0.49) | -0.98 (-1.7, -0.29) | -0.20 (-0.92, 0.49) | -1.4 (-2.0, -0.87) | -2.9 (-3.5, -2.3) | -2.6 (-3.6, -1.7) | 0.07 (-0.10, 0.23) | -0.02 (-0.17, 0.13) |
| Aeromonas | -1.0 (-1.8, -0.22) | -0.93 (-1.8, -0.10) | -1.4 (-2.3, -0.49) | -0.76 (-1.5, -0.04) | -2.4 (-3.2, -1.7) | -3.5 (-4.7, -2.3) | 0.12 (-0.09, 0.33) | -0.10 (-0.29, 0.09) |
| Astrovirus | -0.4 (-1.9, 0.95) | -1.3 (-2.9, 0.34) | 0.06 (-1.6, 1.6) | -3.3 (-4.8, -2.0) | -4.0 (-5.6, -2.5) | NA | 0.21 (-0.23, 0.68) | 0.06 (-0.33, 0.44) |
| Cryptosporidium | 2.0 (1.1, 2.8) | -0.78 (-1.8, 0.21) | 1.4 (0.37, 2.3) | -0.55 (-1.4, 0.27) | 0.12 (-0.75, 0.98) | 1.5 (0.15, 2.8) | 0.05 (-0.19, 0.29) | 0.27 (0.05, 0.50) |
| E. coli O157:H7 | -0.56 (-2.4, 0.54) | -2.4 (-4.2, -1.3) | -2.3 (-4.1, -1.1) | -1.9 (-3.1, -1.1) | -2.9 (-4.7, -1.7) | NA | 0 (-0.42, 0.40) | 0.07 (-0.24, 0.38) |
| E. histolytica | -1.7 (-3.3, -0.37) | -2.4 (-3.8, -1.2) | -0.92 (-2.1, 0.29) | -3.5 (-4.9, -2.4) | -4.0 (-5.7, -2.7) | NA | -0.88 (-1.7, -0.17) | -0.60 (-1.1, -0.18) |
| EAEC | -1.2 (-2.3, -0.22) | -1.1 (-2.3, 0) | -1.3 (-2.5, -0.13) | -2.0 (-3.0, -1.0) | -4.4 (-5.6, -3.3) | -4.4 (-6.4, -2.6) | 0.18 (-0.13, 0.49) | -0.07 (-0.35, 0.21) |
| EPEC | -0.76 (-1.8, 0.25) | -1.3 (-2.4, -0.15) | -1.6 (-2.8, -0.48) | -2.5 (-3.5, -1.5) | -4.0 (-5.2, -2.9) | -4.0 (-5.8, -2.3) | 0.17 (-0.12, 0.47) | -0.03 (-0.3, 0.23) |
| ETEC | -0.95 (-2.2, 0.11) | -2.4 (-3.8, -1.2) | -1.1 (-2.3, 0.19) | -3.0 (-4.0, -1.9) | -4.1 (-5.4, -2.9) | NA | 0.31 (-0.05, 0.66) | 0.49 (0.18, 0.82) |
| Giardia | 1.1 (0.4, 1.8) | -1.3 (-2.0, -0.46) | -1.1 (-1.9, -0.26) | -2.5 (-3.3, -1.9) | -3.4 (-4.4, -2.5) | -3.4 (-5.1, -2.1) | 0.14 (-0.10, 0.37) | -0.02 (-0.23, 0.18) |
| norovirus | 1.8 (1.1, 2.4) | -0.3 (-1.1, 0.46) | 0.53 (-0.23, 1.3) | -1.6 (-2.2, -0.91) | -1.7 (-2.4, -0.99) | NA | -0.02 (-0.23, 0.21) | -0.01 (-0.19, 0.17) |
| rotavirus | -0.33 (-1.9, 1.2) | -0.86 (-2.6, 0.84) | 1.0 (-0.67, 2.8) | -2.2 (-3.7, -0.78) | -2.6 (-4.2, -1.1) | -4.5 (-7.7, -1.7) | 0.23 (-0.23, 0.71) | 0.07 (-0.32, 0.45) |
| sapovirus | -0.05 (-1.6, 1.5) | -0.9 (-2.7, 0.82) | 0.49 (-1.3, 2.3) | -3.5 (-5.1, -1.9) | -5.4 (-7.6, -3.5) | -5.1 (-8.4, -2.2) | 0.08 (-0.44, 0.59) | 0.37 (-0.07, 0.84) |
| Shigella/EIEC | -1.4 (-2.3, -0.60) | -1.9 (-2.8, -1.1) | -1.5 (-2.4, -0.69) | -2.6 (-3.3, -1.9) | -4.4 (-5.9, -3.3) | NA | 0.33 (0.10, 0.59) | -0.15 (-0.39, 0.08) |
| STEC | -2.8 (-6.5, -0.34) | -2.4 (-5.0, -0.22) | -2.3 (-5.0, 0.0) | -3.5 (-6.1, -1.5) | NA | NA | 0.25 (-0.45, 0.97) | 0.49 (-0.24, 1.38) |
| Vibrio spp. | -1.6 (-2.8, -0.55) | -1.4 (-2.5, -0.36) | -2.2 (-3.3, -1.1) | -0.90 (-1.8, 0.0) | -2.3 (-3.2, -1.3) | -2.1 (-3.5, -0.64) | 0.2 (-0.07, 0.47) | 0.25 (-0.01, 0.53) |

Note: There was insufficient data to run the model for A. duodenale, A. lumbricoides, A. lumbricoides, Avian 16S, C. difficile, C. jejuni/coli, canine mtDNA, H. pylori, HIV, Leptospira, M. tuberculosis, N. americanus, Salmonella, SARS-CoV-2, T. trichiura, Toxocara, and zika virus. NA indicates the target was not detected in this matrix. Abbreviations: enteroaggregative E. coli (EAEC), enteropathogenic E. coli (EPEC), enterotoxigenic E. coli (ETEC), enteroinvasive E. coli (EIEC), shiga-toxin producing E. coli (STEC), mitochondrial DNA (mtDNA), Class 1 Integron-Integrase Gene (intl1).

A one standard deviation increase in mean daily temperature (°C) was associated with a small but significant increase in pathogens (0.09 $\log_{10}$, [0.03, 0.15]) (Table 2). This relationship was not significant for bacterial or protozoan pathogens; however, viral pathogens were was associated with increased mean daily temperature (0.19 $\log_{10}$, [0.04, 0.33]). While few point estimates for individual targets were significantly associated with temperature, *Cryptosporidium* (0.27 $\log_{10}$, [0.05, 0.50]) and ETEC (0.49 $\log_{10}$, [0.18, 0.82]) were positively associated with increasing temperature.

There were significant differences in the impact of meteorological variables on pathogen concentrations in specific environmental matrices (S4 Table). Most notably, a one standard deviation increase in 7-day cumulative precipitation was associated with a reduction in pathogens (-0.42 $\log_{10}$, [-0.73, -0.13]) in river water (i.e., dilution), while increasing precipitation was associated with an increase in pathogens (0.26 $\log_{10}$, [0.16, 0.37]) in open drains (i.e., concentration).

### Environmental hazards

Most fecal waste in Maputo is not safely managed and poses an environmental hazard. Based on Maputo's SFD we estimated that 75% of pathogens in WWTP influent and 52% of fecal sludge are safely managed in Maputo (Fig 3 and S3 Text) because a portion of these matrices are treated at the WWTP. There is no treatment of pathogens in wastewater surface discharge, WWTP effluent, open drains, impacted river water, and stormwater and we estimated 0% of the pathogens in these matrices were safely managed.

### Sample size calculations

We applied a basic unadjusted linear regression simulation to estimate the sample size needed to observe a 0.50 $\log_{10}$ reduction (80% power, $\alpha = 0.05$) in the concentration of culturable *E. coli*, human mtDNA, EAEC, *Giardia* and norovirus using the variance observed in our empirical data (S4 Text). We found substantial heterogeneity in the required sample size to observe this effect size. Fewer than 20 samples (e.g., 10 before and 10 after samples) would provide >80% power to observe a 0.50 $\log_{10}$ reduction in many matrix-target combinations, including culturable *E. coli* in WWTP Influent, WWTP effluent, and impacted river water downstream of the WWTP, as well as human mtDNA, EAEC, and norovirus in WWTP effluent (S4 Text). Though several matrix-target combinations with higher variance would requires >250 samples to observe this effect size.

### Discussion

Analysis of wastewater via WBE provides public health insights for communities served by piped sewage systems, but 3.4 billion people rely on onsite sanitation systems globally [54–56]. Tools and lessons learned from wastewater surveillance can and should be broadly applied to other matrices containing excreta, given the concentration of disease burdens in places that are also underserved by conventional wastewater [13]. Although we detected many pathogens less frequently and often at lower concentrations in non-conventional matrices compared to WWTP influent, the data we generated are sufficiently informative for ES in non-sewered communities. Tools and lessons learned from wastewater surveillance can and should be broadly applied to other matrices containing excreta [13]. Unsewered communities in LMICs, including high population density informal settlements, have the greatest burden of disease and may be most at risk of emerging and re-emerging public health threats [57,58]. ES programs, including proposed global WBE efforts [59,60], would benefit from inclusion of non-traditional matrices and enable equitable surveillance of disease activity in the highest risk locations that is independent of socioeconomic status or healthcare access. New methods and systems for environmental surveillance are needed in low-income settings where populations suffer from the greatest burden of disease.

The application of multi-parallel RT-qPCR in this analysis suggests broad utility of casting a wide net for pathogens of interest in settings with few existing data; such efforts may serve as an important screening step when pathogen hazards may not be well characterized or where existing surveillance systems may not capture circulating pathogens. While having a higher lower limit of detection (LLOD) compared with dPCR, multi-parallel RT-qPCR allows for targeted detection and

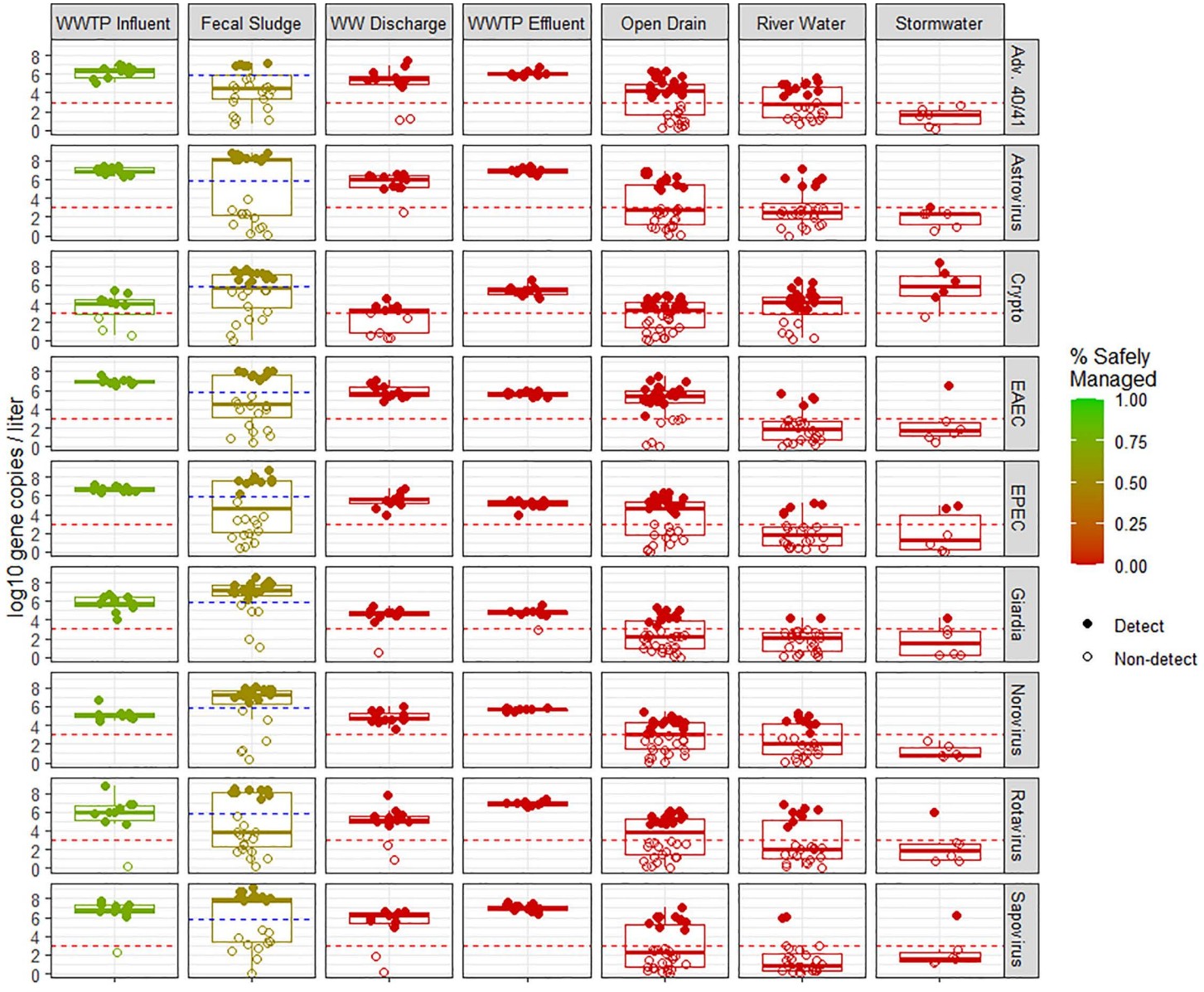

**Fig 3. Concentrations of commonly detected pathogens in environmental matrices.** Note: Dotted line reflect the limit of detection (LOD). Points below the LOD were randomly imputed from 0 to the LOD for visualization purposes and are visualized as non-filled circles. The red to green color scale indicates percentage of pathogens safely managed, which was estimated from Maputo's excreta flow diagram (S2 Text) [25]. Abbreviations: Wastewater treatment plant (WWTP); Wastewater (WW); WW Surface Discharge (WW discharge); adenovirus 40/41 (Adv. 40/41); Cryptosporidium (Crypto); Entero-aggregative E. coli (EAEC); Enteropathogenic E. coli (EPEC).

quantification of dozens of specific pathogens of interest by well characterized and validated assays in a single platform [42]. Given the relatively high concentration of many pathogen targets in wastewaters and other matrices in high-burden settings, these methods may be appropriate; because of the high LLOD these estimates should be considered conservative in terms of prevalence of detections across the matrices we included.

Several of the matrices we assessed are exposure-relevant which may differ from ES in high-income countries with more developed and functional sanitation infrastructure. Quantitative pathogen and flow rate data could be mapped on the

SFD to better visualize environmental hazards with potential for human exposure. These data could inform neighborhood or city-wide infrastructure improvements. Brouwer *et al*. 2024 demonstrated that increasing community coverage had the greatest impact on sanitation effectiveness and that realizing health gains will require consideration of transmission pathways beyond the scope of traditional domestic exposures [61].

However, exposures to enteric pathogens in the environment may vary substantially between settings and certain exposures may only be relevant to specific sub-populations. For example, low-lying neighborhoods may be at greater risk of seasonal flooding compared to neighborhoods at higher elevations. While we observed lower concentrations of pathogens in stormwater compared to other matrices, exposures to stormwater may be greater than other matrices for certain groups. In addition, the use of partially treated or untreated wastewater in irrigation is growing worldwide due to increasing urbanization and climate change associated water stress [62]. Despite the high concentration of pathogens that we observed in wastewater surface discharge, exposure risk from discharge into ocean environments is likely lower compared to wastewater used in agricultural irrigation, as is common downstream of Maputo's WWTP along the Infulene River. Our study was limited to a single rainy and dry season, but there may be seasonal dynamics in enteric infection, shedding, and environmental persistence that impact potential exposures and could not be captured in our single year study [63].

We observed substantial differences in target concentration and variance between environmental matrices. Target presence in fecal sludge was more stochastic than in wastewater, and some fecal sludge samples had higher pathogen concentrations than wastewater influent. Mean target concentrations were lower in the other matrices assessed compared to influent. Environmental matrices that aggregated fecal waste from larger populations (e.g., wastewater influent) had lower variance in their signal than matrices receiving waste from smaller populations (e.g., open drains).

Polio surveillance efforts have encountered a similar challenge in signal variance; surveillance guidelines prioritize wastewater surveillance over other matrices such as open drains or canals but acknowledge that matrices other than wastewater have enabled successful detection of wild poliovirus circulation [64,65]. In communities where wastewater influent is not available similar sensitivity to pathogens in wastewater may be feasible for open drains and impacted river water if more samples are analyzed. Alternatively, lower variance in pathogen signals – and subsequently higher sensitivity – could be achieved by increasing the equivalent sample volume [66].

Our results demonstrate that environmental surveillance efforts should prioritize matrices with the highest and most stable concentrations of fecal contamination that best represent the community of interest. Across pathogen classes, multi-target and individual estimates were most similar between wastewater influent and fecal sludge, suggesting that fecal sludge – particularly composite samples from holding tanks or sludge discharged at treatment plants – may offer the greatest statistical power for surveillance. Pathogens were also frequently detected in open drains and river water, but the substantially higher variance in these matrices indicates that a larger number of samples would be required to achieve statistical confidence comparable to that of wastewater influent or fecal sludge [67].

The utility of fecal sludge as a surveillance matrix depends strongly on the structure of the local fecal sludge disposal chain. In many high-density informal settlements, formal pit emptying is infrequent, and sludge is instead buried onsite or removed manually by informal emptiers [68]. In such contexts, the fecal sludge arriving at a treatment facility may disproportionately represent higher-income neighborhoods, reducing its representativeness for the broader population and limiting its applicability for surveillance. As the burden of disease is often highest in the communities least likely to have their onsite sanitation systems emptied and transported to a treatment plant, methods would be needed to ensure these areas are included in surveillance systems [68]. Potential approaches include direct sampling of fecal sludge from onsite sanitation systems [9,10], or sampling coprophagous flies which feed on fecal waste in these systems [67].

Health impact trials of WASH interventions typically cost millions of dollars and require several years to conduct [57,69–71]. Our simple sample size calculations demonstrate that some engineering controls could plausibly be assessed with tens or hundreds of environmental samples, instead of thousands of clinical samples. Screening different engineering

controls for reductions in environmental fecal contamination would enable the field to "fail faster" and advance the most promising ideas forward to larger, more expensive impact studies [72].

We previously demonstrated that the rank order prevalence of pathogens in fecal sludges was nearly identical to the pathogens detected in feces of people using latrines [9]. Likewise, many of the pathogens we detected in this work commonly infect children in this setting (S5 Table). In low-income urban communities in Blantyre and Mzuzu, we found that fecal sludge depth within a latrine had no impact on pathogen detection, and concluded that sampling from the surface was logistically easiest [10]. Refinement of these methods and those for other types of environmental samples, and their linkage to a broader environmental surveillance system, in conjunction with environmental laboratory systems, are needed to generate actionable public health data in low-resource settings. We describe a few compelling use cases in S6 Table. We can expand efforts to detect and enumerate pathogens that cause diseases with known etiologies (e.g., diarrheal diseases and other neglected tropical diseases, sexually transmitted infections, arboviruses) and inform methods for their control, including emerging and re-emerging diseases and antimicrobial resistance [7,15,73–76]. These methods can also support routine vaccination campaigns (e.g., SARS-CoV-2, rotavirus, poliovirus, *Salmonella* Typhi) [77]. Vaccination may be most important as vaccination programs face new headwinds in the context of misinformation and disinformation about their public health benefits [78].

In urban environments, drainage infrastructure primarily receives runoff from densely populated anthropogenic surfaces such as residential, commercial, and industrial areas, whereas rivers and streams receive runoff from a much broader range of land uses. These differences in hydrology and catchment characteristics may explain why precipitation increased pathogen concentrations in open drains but diluted concentrations in river water. When examined by pathogen class, we observed the same positive association between precipitation and pathogen concentrations for the bacterial pathogens but not for protozoa or viruses. This discrepancy may reflect differences in fate and transport among pathogen types, including variation in attachment to particulates, settling behavior, and environmental persistence [23]. Notably, the dilution of fecal contamination in river water following rainfall contrasts with findings from many high-income settings, where rainfall is typically associated with increased fecal contamination of surface waters [79–81].

We observed similar concentrations of *E. coli* in open drains in Maputo – where sanitation coverage is > 99% – compared to a neighborhood in Accra, Ghana with 48% coverage [82]. Several samples from individual open drains in Maputo had *E. coli* and pathogen concentrations higher than any wastewater effluent sample. Although fecal sludge management is improving worldwide, a 2020 review generated SFDs for 39 cities in low- and middle-income countries and estimated that >50% of human waste was not safely managed [83]. In low-income cities open drains commonly receive residential, commercial, and industrial fecal waste via leaking onsite sanitation, straight piping or dumping [83]. We commonly observed suspected straight pipe connections discharging fecal waste into open drains during field work. Infrastructure aiming to dramatically reduce environmental fecal contamination in Maputo and similar contexts will need to target point sources such as wastewater effluent as well as non-point sources such as fecal waste dumping and straight piping into open drains.

Our results should be considered alongside some important limitations. Lower limits of detection are a result of the volume of sample collected, the concentration methods used, and the downstream analyses (i.e., the equivalent sample volume). Previous viral recovery experiments with BMFS achieved approximately 100% recovery, which supports our direct comparison between BMFS and grab samples of fecal sludge [84]. However, it is possible the recovery of bacteria, protozoa, and helminths via BMFS was less than 100%. The filters we used have an average pore size of 2–3μm. This may have prevented larger pathogens such as *Giardia* cysts (~8x12 μm) and helminth ova (~25x50 μm) from passing through the filter during the elution step, or alternatively the pores may have been too large to recover 100% of viral particles. It is possible we underestimated the prevalence and concentration of these targets. BMFS was designed for logistical feasibility in low-resource contexts, but alternative approaches such as dead-end ultrafiltration may allow for concentration of greater volumes, potentially yielding lower limits of detection [85]. While we determined concentrations of targets of interest in aquatic environments,

we did not collect the volumetric flow rate data that would have enabled comparisons of flow. We decided not to collect flow rate data because it was deemed unsafe or logistically infeasible at several sampling locations. We collected soil and water samples and normalization of these matrices for comparison is not straightforward. We did not attempt a direct comparison of these matrices in our statistical analyses as a result. The methods we used were highly sensitive, enabled volumetric normalization, and quantified a wide range of targets. However, the high cost of BMFS (~$95/sample in 2024) and RT-qPCR via TAC (~$100/sample in 2024) may be cost-prohibitive and limit the scalability of our approach. Lower cost approaches for sampling (e.g., membrane filtration or passive sampling via Moore Swabs) and downstream analysis (e.g., quantification of fewer targets) may be better alternatives for scaling ES in low-resource settings [86]. Finally, some of the assays we selected for environmental monitoring were validated on human fecal specimens. The *Vibrio* spp. assay that we used – which targets the *hylA* gene – was previously published as a *Vibrio cholerae* assay, but cross reacts with other *Vibrio* species [42].

Our results provide evidence of the potential for environmental pathogen surveillance at scale in cities without universal conventional wastewater infrastructure. Recent advances in ES approaches developed in high-income countries can be applied to LMIC contexts but will require adaptation to the local sanitation infrastructure and the intended use case. Further refinement and deployment of these methods in low-income communities would enable ES to generate data in settings where it can have the greatest impact.

## Supporting information

**S1 Fig. Detailed map.**
(PDF)

**S2 Fig. Maputo SFD.**
(PDF)

**S3 Fig. BMFS sample collection.**
(PDF)

**S4 Fig. Fecal sludge collection.**
(PDF)

**S5 Fig. Waste bins.**
(PDF)

**S1 Table. Assays.**
(PDF)

**S2 Table. Standard curve.**
(PDF)

**S3 Table. MIQE checklist.**
(PDF)

**S4 Table. Precipitation and temperature.**
(PDF)

**S5 Table. Pediatric infection comparison.**
(PDF)

**S6 Table. Use cases.**
(PDF)

**S1 Text. Inclusivity in global research.**
(PDF)

**S2 Text. Detailed methods.**
(PDF)

**S3 Text. Pathogen flow table.**
(PDF)

**S4 Text. Sample size calculation.**
(PDF)

## Author contributions

**Conceptualization:** Joe Brown.

**Data curation:** Drew Capone, Victoria Cumbane, Amanda Lai.

**Formal analysis:** Drew Capone, Victoria Cumbane, Jack Dalton, David Holcomb.

**Funding acquisition:** Edna Viegas, Joe Brown.

**Investigation:** Drew Capone, Marcia Chiluvane, Victoria Cumbane, David Holcomb, Erin Kowalsky, Amanda Lai, Gouthami Rao.

**Methodology:** Drew Capone, Marcia Chiluvane, David Holcomb, Gouthami Rao.

**Project administration:** Drew Capone, Erin Kowalsky, Amanda Lai, Elly Mataveia, Vanessa Monteiro, Gouthami Rao, Edna Viegas.

**Supervision:** Elly Mataveia, Vanessa Monteiro, Joe Brown.

**Writing – original draft:** Drew Capone.

**Writing – review & editing:** Marcia Chiluvane, David Holcomb, Erin Kowalsky, Vanessa Monteiro, Gouthami Rao.

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
