## [Decision Letter · Decision Letter 0]

25 Nov 2025

PGPH-D-25-01911

Environmental pathogen surveillance in cities without universal conventional wastewater infrastructure

Dear Dr. Drew Capone,

Thank you for submitting your manuscript to PLOS Global Public Health. After careful consideration, we feel that it has merit but does not fully meet PLOS Global Public Health’s publication criteria as it currently stands. Therefore, we invite you to submit a revised version of the manuscript that addresses the points raised during the review process.

We look forward to receiving your revised manuscript.

Kind regards,

Tintu Varghese, MD

Academic Editor

Journal Requirements:

Please include a complete copy of PLOS’ questionnaire on inclusivity in global research in your revised manuscript. Our policy for research in this area aims to improve transparency in the reporting of research performed outside of researchers’ own country or community. The policy applies to researchers who have travelled to a different country to conduct research, research with Indigenous populations or their lands, and research on cultural artefacts. The questionnaire can also be requested at the journal’s discretion for any other submissions, even if these conditions are not met. Please find more information on the policy and a link to download a blank copy of the questionnaire here: https://journals.plos.org/globalpublichealth/s/best-practices-in-research-reporting. Please upload a completed version of your questionnaire as Supporting Information when you resubmit your manuscript.

Additional Editor Comments (if provided):

Thank you for your submission to the journal. Your manuscript is well written and addresses an important research gap. After careful consideration of the reviewers’ feedback, we request that you revise the manuscript to address the comments provided, particularly those highlighted by Reviewer 2 (see below).

Reviewers' comments:

Reviewer's Responses to Questions

**Comments to the Author**

1. Does this manuscript meet PLOS Global Public Health’s publication criteria? Is the manuscript technically sound, and do the data support the conclusions? The manuscript must describe methodologically and ethically rigorous research with conclusions that are appropriately drawn based on the data presented.? Is the manuscript technically sound, and do the data support the conclusions? The manuscript must describe methodologically and ethically rigorous research with conclusions that are appropriately drawn based on the data presented.

Reviewer #1: Yes

Reviewer #2: Partly

2. Has the statistical analysis been performed appropriately and rigorously?

Reviewer #1: Yes

Reviewer #2: I don't know

3. Have the authors made all data underlying the findings in their manuscript fully available (please refer to the Data Availability Statement at the start of the manuscript PDF file)?

The PLOS Data policy requires authors to make all data underlying the findings described in their manuscript fully available without restriction, with rare exception. The data should be provided as part of the manuscript or its supporting information, or deposited to a public repository. For example, in addition to summary statistics, the data points behind means, medians and variance measures should be available. If there are restrictions on publicly sharing data—e.g. participant privacy or use of data from a third party—those must be specified.requires authors to make all data underlying the findings described in their manuscript fully available without restriction, with rare exception. The data should be provided as part of the manuscript or its supporting information, or deposited to a public repository. For example, in addition to summary statistics, the data points behind means, medians and variance measures should be available. If there are restrictions on publicly sharing data—e.g. participant privacy or use of data from a third party—those must be specified.

Reviewer #1: No

Reviewer #2: Yes

4. Is the manuscript presented in an intelligible fashion and written in standard English?

Reviewer #1: Yes

Reviewer #2: Yes

Reviewer #1: I have reviewed the manuscript titled "Environmental pathogen surveillance in cities without universal convetional wastewater infrastructure". The manuscript explores the utility of using the wastewater effluents to detect pathogens and form pubilc health response.

The work is done in the Maputo region of Mozambique, which is a representative of sewage system present in many cities across the developing world. The work is highly relevant and the manuscript is written in a form that is easy to follow.

I have no comments and the manuscript is recommended for acceptance.

Reviewer #2: The article is well written, and the authors have described a unique approach to analyze suitability of wastewater matrices by pooling quantitative pathogen detection from different matrices and comparing to the standard of WWTP influents commonly used to predict disease prevalence or outbreaks in high-income countries.

However, there are a few critical gaps in the manuscript, that in its current form fails to explain the intent of the authors especially with regards to the study design, and thorough interpretation of results.

1. The authors state in the introduction 3 objectives of which the second was “ Our other goals were to assess presence, concentration, and variance of enteric pathogens in sample types to inform sample size

calculations for potential future use cases, to estimate the fraction of pathogens associated with

uncontained excreta”. They have not sufficiently described why they wanted to carry this out, and if the study design that they had - “purposive sampling” was sufficient to fulfil this objective.

2. The study design should be described in context with the stated objectives - why did they decide on purposive sampling and a justification of why they believe this would be sufficient to address their objectives.

3. The analysis method is fairly well described, however a few more details need to be explained - how where the pathogens weighted in the pooled analysis. Viruses are more easily detected by molecular analysis than bacteria - were the weights based on abundance with more abundant pathogens given less weightage, or potential disease severity?

4. The discussion in the manuscript fails to interpret the results thoroughly and reads like another introduction with recommendations that are not fully supported based on findings from this work. The results from the study based on the mean log difference are not discussed adequately. There is very little discussion on the content and suitability of each matrix for specific pathogens.

5. The results from objective 3 - “fraction of pathogens in uncontained excreta can also be better discussed”. For example, do the results mirror known disease prevalences from Maputo?

5. The most interesting finding was the increase in detection of bacteria and parasites with increased precipitation- other literature elsewhere (including surveillance in drainage systems) would indicate the opposite due to dilution effects. The authors have addressed this in the discussion and have pointed to possible anthropogenic run-offs that lead to increased detection. Why was a similar increase not seen for viruses or parasites detected? Some more elaboration on this would add much more value to the manuscript.

**Do you want your identity to be public for this peer review?** For information about this choice, including consent withdrawal, please see our Privacy Policy..

Reviewer #1: **Yes:** Ankur JamwalAnkur JamwalAnkur JamwalAnkur Jamwal

Reviewer #2: No

---

## [Decision Letter · Decision Letter 1]

26 Feb 2026

PGPH-D-25-01911R1

Environmental pathogen surveillance in cities without universal conventional wastewater infrastructure

Dear Dr. Capone,

Thank you for submitting your manuscript to PLOS Global Public Health. After careful consideration, we feel that it has merit but does not fully meet PLOS Global Public Health’s publication criteria as it currently stands. Therefore, we invite you to submit a revised version of the manuscript that addresses the points raised during the review process.

We look forward to receiving your revised manuscript.

Kind regards,

Tintu Varghese

Academic Editor

Journal Requirements:

Additional Editor Comments (if provided):

Reviewers' comments:

Reviewer's Responses to Questions

**Comments to the Author**

Reviewer #2: All comments have been addressed

Reviewer #3: (No Response)

Reviewer #4: (No Response)

publication criteria? Is the manuscript technically sound, and do the data support the conclusions? The manuscript must describe methodologically and ethically rigorous research with conclusions that are appropriately drawn based on the data presented.? Is the manuscript technically sound, and do the data support the conclusions? The manuscript must describe methodologically and ethically rigorous research with conclusions that are appropriately drawn based on the data presented.

Reviewer #2: Yes

Reviewer #3: Partly

Reviewer #4: Partly

3. Has the statistical analysis been performed appropriately and rigorously?

Reviewer #2: I don't know

Reviewer #3: I don't know

Reviewer #4: Yes

4. Have the authors made all data underlying the findings in their manuscript fully available (please refer to the Data Availability Statement at the start of the manuscript PDF file)?

The PLOS Data policy requires authors to make all data underlying the findings described in their manuscript fully available without restriction, with rare exception. The data should be provided as part of the manuscript or its supporting information, or deposited to a public repository. For example, in addition to summary statistics, the data points behind means, medians and variance measures should be available. If there are restrictions on publicly sharing data—e.g. participant privacy or use of data from a third party—those must be specified.requires authors to make all data underlying the findings described in their manuscript fully available without restriction, with rare exception. The data should be provided as part of the manuscript or its supporting information, or deposited to a public repository. For example, in addition to summary statistics, the data points behind means, medians and variance measures should be available. If there are restrictions on publicly sharing data—e.g. participant privacy or use of data from a third party—those must be specified.

Reviewer #2: Yes

Reviewer #3: (No Response)

Reviewer #4: Yes

5. Is the manuscript presented in an intelligible fashion and written in standard English?

Reviewer #2: Yes

Reviewer #3: Yes

Reviewer #4: No

Reviewer #2: (No Response)

Reviewer #3: Review of "Environmental pathogen surveillance in cities without universal conventional wastewater infrastructure" by Capone et al.

The authors describe a study analyzing environmental samples from different points across the wastewater and sanitation system in Maputo for human pathogens. They further make claims about the potential use of environmental surveillance in public health in areas without the type of wastewater infrastructure common in the global north. The manuscript is generally well written and the analysis of concentrations of different pathogens in different matrices very relevant and well performed. I have reservations on the claims of public health use, given that most of these claims are based on hypotheses and speculation not underlined by the scientific results of this study. The paper reads like a mix of an important environmental study on pathogen presence and concentrations, and an opinion piece of potential uses of such systems at larger scale. I would strongly recommend to separate these two parts into two manuscripts, to achieve clarity as to what is a scientific result and what is an opinion, hypothesis or claim of potential public health benefit.

Besides this general comment I have several major and minor comments:

Major:

1. "conventional" in the title sounds like piped sewer systems with WWTP are the norm globally. However, I do think that unfortunately that is only the case in the global north. Another choice of wording could better reflect this aspect.

2. This is a study conducted in Mozambique, with samples collected in Mozambique, and parts of the laboratory analysis performed in the US. I am surprised that the first and the last authors are both from and based in the global north. What was the role of local collaborators in designing and performing this study and analysis? Wouldn't their contribution warrant first or senior (co-)authorship? What was their role in interpreting results and discussing potential benefits, given that they are probably more familiar with the local situation?

3. The introduction states that wastewater surveillance emerged only after the emergence of COVID-19. However, it was already commonplace for specific diseases (e.g. polio) or specific places.

4. The introduction states some of the use cases of wastewater surveillance in sewered systems, that include estimate of burden, infection trends, spatial and temporal transmission patterns. It then benefits may be greatest in LMICs. However, all the above use cases depend on a piped and centralized system with few sampling locations that are representative of entire neighborhood and city populations and allow for space- and time resolution, which can not easily be achieved in LMIC settings depending decentralized and often informal wastewater and sewage disposal. It would be much preferrable to state potential use cases that actually are realistic in typical LMIC settings.

5. The presented methods are inherently biased towards sampling parts of the city with more formal wastewater or sewage collection, which will be neighborhoods that tend to have a better socioeconomical level. The more deprived a subpopulation, the more there will be a tendency to rely on informal disposal mechanisms or seal septic pits and dig new latrines. On the other hand, in the populations the most at risk of infectious diseases will often live in these neighborhoods and will thus be underrepresented in the environmental surveillance approach presented here. This is a very important limitation that needs to be discussed in depth.

6. Sample size calculation: 0.5 log10 is a factor of about 3.16, which is a very substantial reduction in concentration, on average over a large city. It would be useful to have sample sizes for more realistic reductions.

7. The comparison detection rates of pathogens in this study vs clinical prevalence studies in table S5 shows high discordance between the two methods. This is a very important result given that the authors state public health benefits. This result should thus not be in the supplement but in the main text, and potential reasons for discordance and biases of this study with respect to population prevalence and clinical surveillance should be discussed in detail.

8. The above probably highlights that the methods applied here have a problem of representativity of the underlying population if not applied with a much denser spatial and temporal sampling, which is very challenging in terms of logistics and thus has important feasibility and scalability constraints that should be discussed.

9. The small number of samples (96 for the entire Maputo in 2 years!) further shows how limited the representativity of this study is and how challenging it would be to extract information on the underlying population level prevalence in space and time beyond simple presence-absence results.

10. Based on the above limitations the claimed potential use cases (introduction, discussion, table S6) need to be much more thoroughly and critically assessed for feasibility, scalability and if they are preferrable to clinical surveillance.

Minor:

1. The definition of stormwater being only from large standing pools seems peculiar.

2. Explain unit MPN

3. Effect of precipitation most likely affects differentially affects different matrices. The authors don't show any results on this.

4. Figure 3: I don't think it's appropriate to visualize point < LOD randomly between 0 and LOD.

5. The authors discuss polio surveillance in wastewater. One important difference here is that for polio surveillance, given the disease is almost eradicated, presence/absence information is sufficient. For more common pathogens representative quantification by population is required to infer relevant public health information.

6. LL 500: But at very different concentrations, and because of overflow of combined sewer systems during intense precipitation.

I didn't have sufficient time and background to assess if the sampling, laboratory and statistical analysis were appropriate and performed well and recommend that these are assessed by reviewers with the appropriate background.

Reviewer #4: PGPH-D-25-01911R1: statistical review

SUMMARY. This study assesses mean differences in molecular target concentrations relative to wastewater influent. The core statistical analysis relies on a battery of censored regression models, to account for data that are left-truncated by a lower detection limit. Regression models are first estimated separately for each target concentration and then pooled according to pathogen-specific classes. Pooled regressions included random effects to account for latent heterogeneity between molecular targets. Methods are correct but they are not clearly explained, see the specific issues below. I also add some further comments that could improve material presentation.

SPECIFIC ISSUES

1. Lines 224- 229. Using terms like “pooled regression” and “weighting schemes” can create some confusion in an audience who is not familiar with Bayesian mixed effects regression. Why don’t the authors simply say that they are using a battery of mixed effects models where intercepts and slopes vary across targets?

2. I guess that the analysis assumes that the log-transformed data are normally distributed. The authors should provide some empitical evidence that this assumption is fulfilled.

3. Statistical reporting (see the guidelines on the journal web site). Papers that rely on Bayesian methods must include details about the choice of priors and how they were selected. Markov chain Monte Carlo settings should also be reported.

4. The R code used for the analysis should be included in the supplementary material. This would help the reader to replicate the analysis.

ADDITIONAL COMMENTS.

1. Figures 2 and 3. Dots below the detection limit should be marked differently (use either a different color or symbol). It should be immediately visible that these points are imputed and not observed.

2. Table 2. Credible intervals where both the lower and the upper limit share the same sign should be marked in a special way (e.g. boldface), to help the reader detect significant differences.

3. The statistical analysis is Bayesian. The term “confidence interval” should be replaced by “credible interval”

**Do you want your identity to be public for this peer review?** For information about this choice, including consent withdrawal, please see our Privacy Policy..

Reviewer #2: No

Reviewer #3: No

Reviewer #4: No

---

## [Editor Report · Decision Letter 2]

18 Mar 2026

Environmental pathogen surveillance in cities without universal piped wastewater infrastructure

PGPH-D-25-01911R2

Dear Drew Capone,

We are pleased to inform you that your manuscript 'Environmental pathogen surveillance in cities without universal piped wastewater infrastructure' has been provisionally accepted for publication in PLOS Global Public Health.

Best regards,

Tintu Varghese, MD

Academic Editor